# Plant Histone HTB (H2B) Variants in Regulating Chromatin Structure and Function

**DOI:** 10.3390/plants9111435

**Published:** 2020-10-25

**Authors:** Janardan Khadka, Anat Pesok, Gideon Grafi

**Affiliations:** French Associates Institute for Agriculture and Biotechnology of Drylands, Jacob Blaustein Institutes for Desert Research, Ben-Gurion University of the Negev, Midreshet Ben Gurion 84990, Israel; janardankhadka@gmail.com (J.K.); anat.pesok@mail.huji.ac.il (A.P.)

**Keywords:** *Arabidopsis*, chromatin organization, flowering, histone H2B (HTB) variants, histone fold, histone modification, nuclear localization, nucleolar retention, nucleosome

## Abstract

Besides chemical modification of histone proteins, chromatin dynamics can be modulated by histone variants. Most organisms possess multiple genes encoding for core histone proteins, which are highly similar in amino acid sequence. The *Arabidopsis thaliana* genome contains 11 genes encoding for histone H2B (HTBs), 13 for H2A (HTAs), 15 for H3 (HTRs), and 8 genes encoding for histone H4 (HFOs). The finding that histone variants may be expressed in specific tissues and/or during specific developmental stages, often displaying specific nuclear localization and involvement in specific nuclear processes suggests that histone variants have evolved to carry out specific functions in regulating chromatin structure and function and might be important for better understanding of growth and development and particularly the response to stress. In this review, we will elaborate on a group of core histone proteins in *Arabidopsis*, namely histone H2B, summarize existing data, and illuminate the potential function of H2B variants in regulating chromatin structure and function in *Arabidopsis thaliana*.

## 1. Introduction

The basic structural unit of chromatin is the nucleosome, which is composed of DNA wrapped around histone octamer made of two of each of the core histone proteins, namely, H2A, H2B, H3, and H4. These histone proteins share a structural motif, the histone fold [1] that mediates the interactions between core histone proteins and between histone octamer and duplex DNA. The amino-terminal histone tail is essentially unstructured and protruding outside the nucleosomal disc, where it can contact nearby nucleosomes as well as multiple proteins that affect the structure of chromatin and its function. The structure of chromatin is highly dynamic, enabling the transition between permissive (euchromatic) and repressive (heterochromatic) chromatin. This dynamic structure is controlled by multiple types of reversible chemical modifications that occur on the DNA (mainly cytosine methylation) or on the core histone proteins (e.g., acetylation, methylation, ubiquitination). These chemical modifications can directly affect the interaction of histones with DNA or generate binding sites for the recruitment of proteins or protein complexes that affect the structure and function of chromatin and, consequently, differentiation and development as well as response to stress [2].

In addition to chemical modification of histone proteins, the dynamic of chromatin and accessibility can be modulated by histone variants. Most organisms possess multiple genes encoding for core histone proteins, which are highly similar in amino acid sequence. For example, the *Arabidopsis* genome contains 11 genes encoding for H2B (designated HTBs), 13 for H2A (HTAs), 15 for H3 (HTRs), and 8 genes encoding for histone H4 (HFOs) [3]. The findings that histone variants may be expressed in specific tissues and/or during specific developmental stages suggest that the histone variants might have specialized functions in regulating chromatin structure and function, which are currently only partly known and might be important for better understanding growth and development and particularly the response to stress. The relationship between histone variants and chromatin structure is well exemplified by the centromeric histone H3 (cenH3; encoded by *HTR12* in *Arabidopsis*), found in all eukaryotes, replacing canonical H3 proteins at centromeric regions and are important for chromosome segregation [4]. Other H3 variants, H3.1 and H3.3, are enriched in repressive and permissive chromatin structures, respectively [5]. While H3.1 variants function as canonical histones which are assembled onto nucleosomes during DNA replication, H3.3 variants act as replacement histones highly associated with gene bodies of transcribed genes [6]. The study of H2A variants in *Arabidopsis* revealed four major types of homotypic nucleosomes containing exclusively H2A, H2A.Z, H2A.X, or H2A.W, which are predicted to regulate chromatin dynamics and function [3,7,8]. The phosphorylated H2A variant H2A.X (γH2A.X) defines regions of damaged DNA, particularly within euchromatin, while phosphorylated H2A.W.7 appears to define DNA damage in heterochromatin [9]. Accordingly, H2A.X variants encoded by *H2A.X.3* (*HTA3*) and *H2A.X.5* (*HTA5*) genes in *Arabidopsis* [7] are characterized by a four-amino-acid carboxy-terminal motif (SQEF) whose serine residues undergo phosphorylation by the Ataxia Telangiectasia-Mutated (ATM) and Ataxia Telangiectasia-Mutated and Rad3-related (ATR) kinases at sites of ionizing radiation-induced double-stranded DNA breaks in preparation for repair [8,10]. Similarly, The SQ motif in the H2A.W.7 variant is phosphorylated in response to DNA damage by ATM [9,10]. Thus, while, in plants, most studies and review articles focus on histone H3 and H2A variants [3,11,12,13,14], the study of histone H2B variants is emerging [15,16], and this topic represents the focus of the present review.

## 2. Animal H2Bs

Histone H2B protein family in human displays extensive coding variation and constitutes the highest number of proteins among the different types of histones with 15 unique variants [17]. The analysis of the H2B family members across different eukaryotic genomes revealed that following gene duplication, H2B variants undergo a rapid process of diversification and functional specialization, which particularly impacts the male germinal cell lineage [18,19]. A study in mice revealed that during spermatogenesis, testis-specific H2A (TH2A) and H2B (TH2B) dimerize to form H3/H4-deficient nucleosomal particles making up the condensed chromatin in spermatids [20]. A detailed study of TH2B function in mouse models whereby TH2B was modified or depleted uncovered its special role in chromatin reorganization, including facilitating the removal of histones during spermatogenesis and the transition from histone-based chromatin structure into nucleoprotamine particles that assist chromatin compaction [21]. Although TH2B was initially identified in mammalian testes, this variant was later found to be highly expressed in oocytes [22]. In mice, the generation of induced pluripotent stem cells (iPSCs) by the four transcription factors OCT4, SOX2, KLF4, and c-MYC is enhanced by the co-expression of the testis-specific variants TH2A and TH2B [23]. Moreover, these two histone variants appear to control the establishment of an open chromatin state following fertilization making the parental genome competent for transcription [22]. The crystal structure of human testis-specific nucleosomes containing TH2B revealed that the TH2B is unable to form the water-mediated hydrogen bonds with H4 Arg78 residue due to a local structural difference around TH2B Ser85. Thus, hydrogen bonding is generated between Asn84 of canonical H2B but not with Ser85 of TH2B, which may facilitate the formation of unstable nucleosome for global histone replacement during spermatogenesis [24].

Recent findings in *Hydractinia* (hydrozoan) revealed that germ cell-specific H2B variants replace canonical H2B in the early stages of spermatogenesis without changing the level of chromatin compaction in the sperms [25]. A new role was described for the mouse-specific H2B.E variant in modulating the life span of odor-sensing neurons that replace canonical H2B in inactive neurons. The H2B.E is very similar to canonical H2B, with only five changes in amino acid sequences [26]. The neurons expressing the H2B.E variant showed a low level of H2BK5-methylation, indicating the functional speciation might be associated with post-translational modification of the H2B variant; mono-methylation of H2BK5 is positively correlated to the transcriptional activity of associated genes [26,27].

## 3. Plant H2Bs

Earlier work on plant histones have demonstrated that plant H2B proteins are less conserved among all other core histone proteins [28,29]. The construction of phylogenetic trees revealed that HTB variants tend to cluster according to species. Thus, the N-terminal tails of HTB variants are more conserved within species than between species [16,30]. Alignment of the amino acid sequences of the *Arabidopsis* HTB variants shows that while the C-terminus is highly conserved (except for HTB8, At1g08170), most differences occur in the N-terminal region ([15,31], see also Figure 1A). Based on a phylogenetic tree (Figure 1B), HTB variants can be divided into three classes. Class I contains HTB1(At1g07790), HTB2 (At5g22880), HTB3 (At2g28720), HTB4 (At5g59910), HTB9 (At3g45980), and HTB11 (At3g46030) and class II HTB5 (At2g37470), HTB6 (At3g53650), HTB7 (At3g09480), and HTB10 (At5g02570). Class I variants can be further divided into class I-A (H2B4, 9, and 11) and class I-B (HTB1, 2, and 3) and class II into class II-A (HTB7 and 10) and class II-B (HTB5 and 6). The third class (class III) contains a single variant, HTB8, which is highly divergent from other HTB proteins. It consists of an extended N-terminal region and has a calculated molecular mass of about 27 kDa, which is significantly higher than other HTBs ranging from 14 to 16.5 kDa. BLAST analysis of HTB8 amino acid sequence against the NCBI database revealed HTB8 orthologs with an extended N-terminus mostly within the eudicot clade of angiosperms and other clades, monocots, and magnoliids. They differ in the length of the N-terminal extension and their molecular weight ranges from ~19 kDa (e.g., *Oryza brachyantha*, H2B.L4-like) to ~32 kDa (e.g., *Sorghum bicolor*, H2B.4). Sequence alignment (Appendix A) followed by the construction of a phylogenetic tree (Appendix A) showed that HTB8-like variants within Brassicales are highly conserved along the protein sequence, including the N-terminal extension (Appendix A) and are clustered according to their lineages within Brassicaceae (Appendix A). Although similarity in protein sequence of the N-terminal extension decreases with increasing phylogenetic distance (Appendix A), HTB8 orthologs in other families were clustered according to their lineages within eudicots (Appendix A). Notably, as indicated previously [15], a conserved motif is identified within the N-terminal extension of HTB8 orthologs in angiosperms (Magnoliids, Monocots, and Eudicots clades) with the core sequence KVVxETVxVxV (Appendix A). This conserved motif points to a common ancestor for HTB8-like proteins in angiosperms, and might be central in determining their function in regulating chromatin organization and dynamics. BLAST analysis of the HTB8 C-terminal portion (122-243) revealed high amino acid sequence similarity with H2B proteins ranging from viridiplantae to metazoan, pointing to functional and structural conservation of the histone fold domain among eukaryotes.

The sequence alignment of class I and class II HTBs showed that class II differs from class I by the lack of 11 (HTB5, HTB6) and 21/19 (HTB7/HTB10) amino acids in the N-terminal region and by the substitution of several amino acids (highlighted green in Figure 1A). These features raised the hypothesis that the deletion (as in class II) and addition (as in H2B.8) of amino acids at the N terminus may determine the functional heterogeneity between class I and class II/III HTB proteins and we will elaborate on these differences and their mechanistic significance.

## 4. Amino Acid Substitutions and Functional Significance

Closer examination of the amino acid sequences of class I and class II HTB variants revealed that some amino acid residues in class I are modified in class II. These include the conversion of glycine (G) 39 to acidic, negatively charged glutamic/aspartic acid (E/D) in class II, a conservative mutation where leucine (L) at position 42 of class I variants has been converted to isoleucine (I) in class II, and serine (S) at position 82 in class I (i.e., HTB4, HTB9) to glycine (G) in class II; serine at this equivalent position is retained in the HTB8 variant. Notably, the conservative mutation of L into I and vice versa might have functional consequences. Although both isomers are non-polar hydrophobic amino acids having the same molecular weight, they differ in the position of a branching methyl group in the side chain, which alters their capacity to interact with other molecules leading to changes in protein properties [33,34]. The conversion of G39 in class I to E/D in class II may affect the properties of the N-terminal tail and its capability to interact with neighboring amino acids (e.g., positively charged K and R) or with the linker DNA or DNA in neighboring nucleosomes. These differences may underlie some of the functional divergences between HTB variants. Accordingly, the conversion of S82 in class I into glycine in class II may have functional relevance in facilitating the formation of repressive chromatin. Protein modeling of class I HTB9 revealed that S82 (GIS**S**) is located at the boundary between loop 1 and α2 helix and based on The Arabidopsis Protein Phosphorylation Site Database (PhosPhAt 4.0), S82 is phosphorylated in most class I HTBs. This phosphorylation might change the properties of the protein and its capacity for interaction with neighboring amino acid residues and with the DNA. Thus, in humans, H2B S57 (GIS**S**), which is equivalent to *Arabidopsis* class I HTB S82 (Figure 2A) and is located similarly at the boundary between L1 and α2, forms hydrogen bonds between the phosphate group of the DNA and the main-chain amide nitrogen atom (yellow arrow in Figure 2C) as well as side-chain hydroxyl oxygen atom (black arrow in Figure 2C) of S57 (Figure 2B,C) [35], which contribute to nucleosome stabilization. However, it is expected that phosphorylation of serine at this position (S82), as occurs in *Arabidopsis* class I HTBs, might disrupt this interaction leading to nucleosome instability that facilitates gene transcription. Thus, the conversion of S82 into Glycine in class II HTBs is expected to stabilize nucleosomes and contribute to the formation of repressive chromatin.

## 5. H2B Variants Posttranslational Modifications

Post-translational modifications of H2B variants isolated from euchromatin-enriched fraction obtained from cultured cells were studied by using reversed-phase chromatography with tandem mass spectrometry [31]. This euchromatic fraction was enriched with class I HTB-variants HTB1, HTB2, HTB4, HTB9 and HTB11 displaying posttranslational modifications including acetylation of lysines 6, 11, 22, 27, 32, 38 and 39, ubiquitination of Lys-145 and methylation of lysines 3 and 11 [31]. Notably, HTB3 was absent from this euchromatic fraction, although it appears to be highly associated with the body of transcribed genes and highly expressed in a non-periodic manner in proliferating cells [15]. A summary of PTMs (Plant PTM Viewer https://www.psb.ugent.be/webtools/ptm-viewer/ptm.php), which were identified experimentally in HTB proteins is given in Appendix A. Of particular interest is the acetylation of K72 (within the α1 helix of the histone fold) in class I HTBs but not in class II. Structural examination showed that this lysine residue is equivalent to K47 of human H2B and is facing outside the nucleosomal disc (Appendix A), where it can be associated with other nucleosomes or more likely with other protein or protein complexes to drive chromatin dynamics. Indeed, histone lysine acetylation has been shown to serve as a binding site for the recruitment of various bromodomains containing proteins, which are often associated with permissive chromatin and gene transcription [36,37]. Thus, it is plausible that acetylation of K72 in class I HTBs might have an important role in recruiting chromatin modifiers that facilitate the formation of transcriptionally active chromatin, further supporting the idea that class I HTBs may be involved in the activation of gene transcription.

Most studies in *Arabidopsis* related to HTB chemical modifications focused on monoubiquitination of lysine 145 (H2BmUb), a modification that in animals, commonly signals for methylation at lysine 4 of histone H3 (H3K4) by a COMPASS complex to bring about active chromatin [38]. In *Arabidopsis,* however, a recent study showed that histone H3K4 trimethylation by a COMPASS-like complex is independent of H2BmUb [39]. H2BmUb in *Arabidopsis* has been implicated in the activation of gene expression [40,41], while deubiquitination of H2BmUb was reported to be required for repressive histone H3 methylation and DNA methylation to bring about heterochromatin and gene silencing [42]. However, the role played by H2B monoubiquitination and deubiquitination in regulating chromatin dynamics is highly complicated. In rice, in the absence of pathogen attack, certain disease-related genes are repressed by the SWI/SNF2 ATPase BRHIS1, which is recruited to the promoter region of these genes via specific interaction with monoubiquitinated H2B.7 and certain H2A variants [43]. In yeast, ubiquitination of H2B followed by deubiquitination of H2BmUb is required for enhancement of transcription of certain genes [44]; the ubiquitination/deubiquitination cycle appears to be a general phenomenon in gene transcription also found in plants and animals [45,46]. In addition, H2BmUb is involved in the regulation of various developmental processes, including cell cycle progression during early leaf and root growth [47], seed dormancy [48], photomorphogenesis [40], flowering time [49,50,51], cutin and wax biosynthesis [52], dynamics of microtubules [53] and in plant defense against necrotrophic fungi and in controlling the expression of disease resistance genes [54,55]. Ubiquitination, particularly polyubiquitination, has been implicated in protein degradation [56], yet, monoubiquitination (mUb), as well as non-destructive polyubiquitination of histone H2B appears to play key roles in regulating chromatin structure and function, DNA damage response, differentiation of stem cells and malignancy [57,58]. The enzymes responsible for H2B ubiquitination have been studied in various organisms ranging from plants to animals. In the yeast *Saccharomyces cerevisiae* H2BmUb is dependent on Rad6 that encodes for the ubiquitin-conjugating enzyme Ubc2; H2BmUb is not detected in *rad6* mutants [59]. Rad6 is acting together with E3 enzyme BRE1 and LGE1, to catalyze the mUb of H2B at lysine 123. Similarly, in plants, the E2 ubiquitin-conjugating enzymes UBC1/UBC2 acting together with the ubiquitin E3 ligases encoded by the *HISTONE MONOUBIQUITINATION1* (*HUB1*) and *HUB*2 to control HTB monoubiquitination. The steady-state level of H2BmUb in *Arabidopsis* is controlled by the interplay between uniquitination and the deubiquitination module (DUBm) that contains three proteins including UBP22, the major H2BmUb deubiquitinase [46]. The DUBm is controlled by the DE-ETIOLATED 1 (DET1), an evolutionary conserved factor of the ubiquitin-proteasome system that destabilizes the DUBm by controlling the degradation of the SGF11 DUBm subunit [60]. The repression of flowering by H2BmUb was linked to the expression of *FLOWERING LOCUS C* (*FLC*) [49], a gene encoding a MADS-box transcription factor that negatively controls flowering via interacting and repressing the FLOWERING LOCUS T (FT), a positive regulator of flowering [61,62]. Loss of H2BmUb in *hub1* and *hub2* mutants or in *ubc1/ubc2* double mutant has led to the suppression of *FLC* gene expression and consequently to early flowering [49,50]. Thus, while mUb of HTB is an important modification controlling flowering in *Arabidopsis*, the specific HTB variant(s) involved in flowering time have not been explored. Notably, all HTB proteins, excluding HTB8, in *Arabidopsis,* share high amino acid sequence identity at their C termini, and the ubiquitination site (K) at position 145 is highly conserved (highlighted yellow in Figure 1A).

## 6. Expression of HTB Variants

The significance of HTB chemical modifications is beginning to be explored, yet, not much is known about the significance of H2B variants for plant growth and development and response to stress. The expression pattern of HTB variants in plant tissues and during various stages of development might give some clues regarding their potential function in regulating chromatin dynamics. The transition from leaf cells to protoplasts following treatment with cell wall degrading enzymes is accompanied by extensive changes in chromatin structure that undergo decondensation followed by recondensation of pericentric heterochromatin [63,64,65] as well as in gene expression profile, including changes in chromatin related genes [66]. Among them are class II H2B variants HTB5, HTB6, and HTB7; class I HTB4 and HTB9 proteins, which are highly expressed in leaves, were down-regulated in dedifferentiating protoplasts (Figure 1C). This might implicate class II HTBs in controlling chromatin dynamics and the interplay between open and close chromatin. Also, the absence of class II variants from the euchromatic fraction reported by Bergmüller et al. [31] may implicate them in establishing and maintaining heterochromatin. The male gamete *HTB8*, as well as HTB10, was reported to be the only *HTB* genes expressed in the *Arabidopsis* pollen [67], though searching transcriptome databases (Genevestigator platform) revealed that other class II *HTB* genes are highly expressed in sperm cells where class I HTBs are essentially silent (Figure 1D and ref. [15]). Notably, while *HTB8* is strongly expressed in pollen and sperm cells, it is mildly expressed in seeds, endosperm, silique, and root protoplasts (Figure 1D). A recent comprehensive analysis of the Arabidopsis HTB genes’ expression pattern showed that *HTB5*, *HTB6,* and *HTB7,* but not *HTB8* and *HTB10,* are essentially expressed in most tissues and organs where class I *HTB*s are expressed [15]. On the other hand, class I *HTB*s were essentially not expressed in sperm cells and pollen where class II/III *HTB* variants are highly expressed; *AtHTB8* also showed high expression in mature seeds and in sperm cells [15]. Furthermore, proteome analysis of nuclear proteins derived from the monocotyledonous *Lilium davidi* male floral organ cells, namely, vegetative cells, generative cells, and sperm cells revealed 8 H2B variants. Two H2B variants appeared to be male germline-specific variants and were designated mgH2B and mgH2B.in [68]. The mgH2B.in variant is 193 amino acids in length has an N-terminal extension with a modified motif, characteristics of HTB8 orthologs, with the sequence of **R**K**V**T**ET**QT**L**K**V** (Appendix A). Considering that chromatin in sperm cells as well as in seeds/caryopses is highly condensed [69,70], it is possible that class II and class III HTBs participate, at least partly, in the establishment and maintenance of repressive chromatin conformation, though we cannot exclude a possible function in facilitating the formation of euchromatin [22].

Notably, a seed-specific H2B group, designated H2B.S, has been suggested for angiosperms, which includes the *Arabidopsis* class III HTB8 as well as H2B variants from tomato (Solyc06g074750.1/K4C9J2), rice (LOC_Os09g39730/HTB713), and maize (GRMZM2G442555) [15]. It should be noted, however, that *HTB8* orthologs are not restrictively expressed in seeds/caryopses but show high expression of the gene in other tissues, particularly in pollen cells [15,30], see also Figure 1D). Thus, in our opinion, the characterization of the HTB8-related group as ‘seed-specific’ should be reexamined.

## 7. Subnuclear Localization and Genomic Distribution of HTB Proteins

Transient expression in protoplasts showed that HTB5-GFP is restrictively localized to the nucleus, often displaying a peculiar localization at the periphery of the nucleolus or punctuated nuclear distribution [71]. The localization at discrete domains around the nucleolus reminiscent the localization of methyl CpG binding domain (MBD) proteins (AtMBD5, AtMBD6) in *Arabidopsis* nuclei, showing a preference for perinucleolar chromocenters [72] and might be involved in silencing the nucleolar organizing region (NOR) [73]. Transgenic plants expressing HTB5-GFP under the 35S promoter displayed nuclear localization in mesophyll cells, guard cells, trichomes, and root cells. Furthermore, protoplasts derived from transgenic 35S-HTB5-GFP showed that HTB5-GFP is mainly found at discrete nuclear sites, which are likely to be chromocenters [71]. The accumulation of HTB5 in chromocenters may suggest involvement in maintaining/establishing heterochromatin, or involvement in a unique, as yet an unknown function at chromocenters of dedifferentiating protoplast cells. Nucleolar localization has been reported for histone H2B in bovine liver cells and chicken erythrocytes [74] and, more recently, in cultured cancer cells [75]. The H2B-EGFP protein was initially accumulated in the nucleolus following ectopic expression in HeLa cells but gradually diminished and incorporated into chromatin [76]. Localization of H2B in the nucleolus has been attributed to the presence of a nucleolar localization/retention signal, which is rich in basic amino acids (KKRKRSRK) [77]. This conserved motif was first characterized in yeast as histone H2B repression domain (HBR) required for repression of many genes located adjacent to telomeres or function in vitamin and carbohydrate metabolism [77]. Similar motifs exist in all classes of the *Arabidopsis* HTB proteins with the core sequence of KKKK k/r/m k/s KK (Appendix A) and are thus predicted to be transiently localized in the nucleolus. Indeed, searching the *Arabidopsis* nucleolar protein database (bioinf.scri.sari.ac.uk/cgi-bin/atnopdb/get-all-data) [78] revealed that among the 217 proteins identified in a proteomic analysis of nucleoli isolated from *Arabidopsis* cell culture, four are histone H2B, namely, HTB2, HTB4, HTB9, and HTB10. In addition, 6 histone H2A, 1 histone H4, and 1 histone H1 were listed in the nucleolar protein database. In mouse cycling cells, serine residue (S32) within this motif (KKRKR**S**RK) is phosphorylated by the ribosomal S6 kinase 2 (RSK2), and this phosphorylation is linked with cell transformation [79]. Nucleolar localization was also reported for H2A in *Brassica oleracea* floral meristem cells as well as in yeast and human cells, where it is methylated at glutamine 105 (Q105) by the fibrillarin methyltransferase [80,81].

A recent study demonstrated the chromatin localization of class I HTB proteins using chromatin immunoprecipitation (ChIP) with specific antibodies followed by high-throughput sequencing (ChIP-seq) [15]. This analysis revealed that HTB3 is particularly enriched in gene bodies of highly transcribed genes, similar to the distribution of the replacement histone H3.3 variants and thus may serve as a replacement variant. It is depleted from constitutive heterochromatin, transposon fragments and from long, intact transposable elements (TEs). HTB1 (H2B.1) and HTB2 (H2B.2) are more abundant in pericentric heterochromatin and are enriched in the 5’ and 3’ regions of poorly transcribed genes. Class I-A HTBs appear to be associated with gene bodies independently of their expression level [15]. Careful examination of the data [15] showed that while histone H3 is highly enriched in the body of TE genes, class I HTB proteins were essentially depleted, suggesting that these TE regions may be occupied by class II/III HTB proteins to facilitate their compaction and silencing.

## 8. Concluding Remarks

We hypothesize, based on their amino acid composition, expression pattern, and structural consideration, that HTB variants in *Arabidopsis* can be divided into two functionally divergent groups, namely, Class I and Classes II/III, whose functions are essentially associated with permissive and repressive chromatin, respectively, during specific stages of plant growth and development. Thus, while class I HTBs carry a major function in regulating chromatin dynamics during growth and development of vegetative tissues, class II/III HTBs play a key role in controlling chromatin structure and function in reproductive tissues. The mechanisms underlying the functional divergence of Class I and Classes II/III are beginning to be explored. This divergence might be related to changes in amino acids at specific positions between class I and class II and their potential post-translational modifications including the conversion of S82 in class I into G82 in class II and the potential acetylation of K72 in class I but not in class II HTBs. The divergence between HTB variants may also be related to the presence (as in HTB8 N extension) or absence of the 11 or 19/21 amino acid motifs in class II, otherwise present in class I variants. We presume that this motif serves as a docking site for the variant function. HTB3, which is mostly expressed in leaves, shoot apical meristem as well as in proliferating cells is appeared to function in transcription occupying the gene body of highly transcribed genes [15] and may be involved in transcriptional elongation. HTB8 has highly diverged from class I and class II HTBs and might represent a special HTB group, which is involved in chromatin organization in male gametes as well as in dormant seeds [15], but its exact function is presently unknown. As mentioned above, specific amino acid substitutions between class I and class II might also be significant for the functional divergence between the two classes and should be in focus in future studies.

## Figures and Tables

**Figure 1 plants-09-01435-f001:**
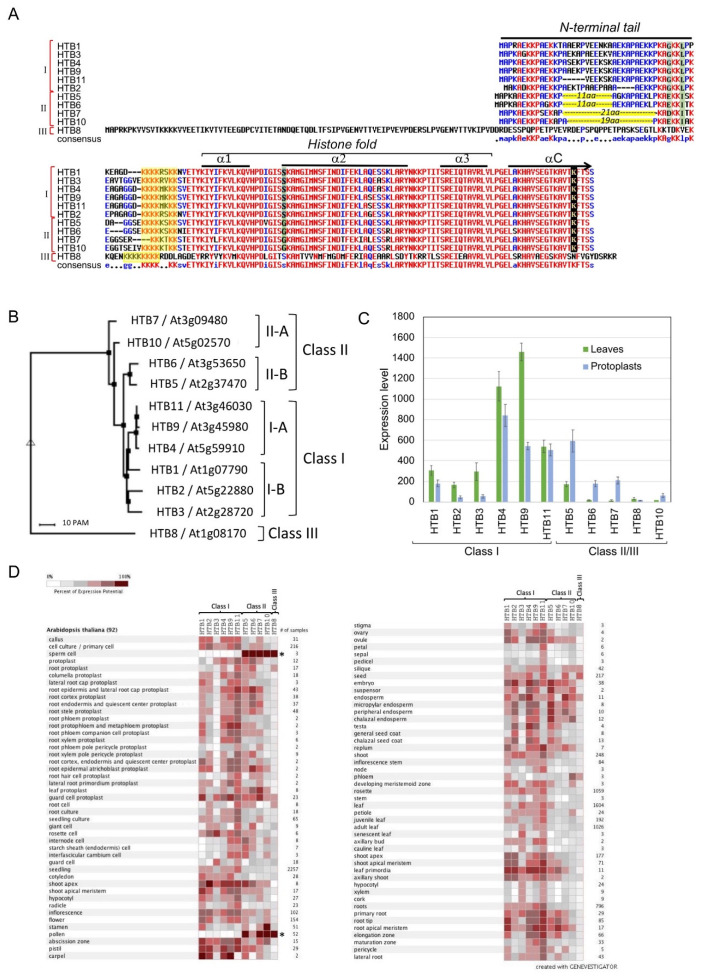
HTB variants in *Arabidopsis thaliana* (HTBs). (**A**) Amino acid sequence alignment of HTB variants in *Arabidopsis thaliana* using Multalin interface page [32]. Variation in amino acids between classes I, II, and III are highlighted green, and the C terminus lysine (K) residue undergoing monoubiquitination is highlighted and marked by an arrow. The yellow box indicates a potential nucleolar localization/retention signal, which is also known as the histone H2B repression domain (HBR). The N-terminal tail and the histone fold region with alpha (α) helices are marked. αC indicates the C-terminal α helix. The position of the 11 and 19/21 aa motifs deleted in class II are highlighted yellow. (**B**) A phylogenetic tree based on the multiple alignments in (A) demonstrating the clustering of H2Bs into three major classes and subclasses. (**C**) Expression pattern of the *Arabidopsis* HTB variants in leaves and dedifferentiating protoplasts [based on 66] (**D**) Expression pattern of the HTB variants in various tissues of *Arabidopsis*. Data obtained using the Genevestigator platform. Note that in sperm cells and pollen (asterisks), Class I expression is declined while that of Class II and class III is strongly elevated.

**Figure 2 plants-09-01435-f002:**
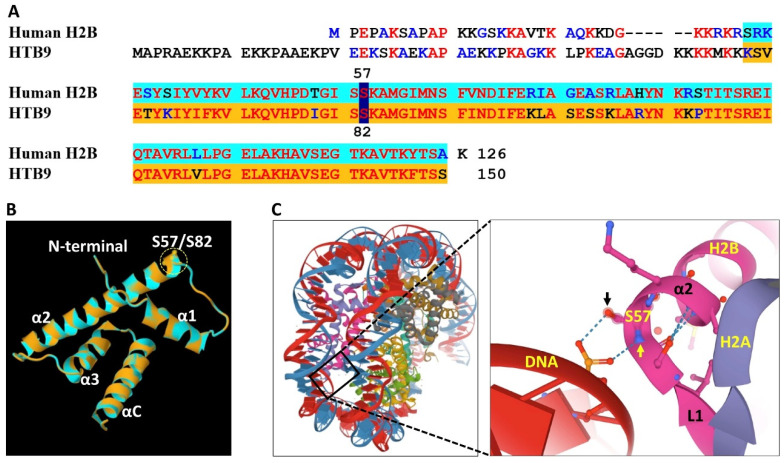
Comparison of H2B protein structure between human and *Arabidopsis*. (**A**) Alignment of amino acid sequences of H2B from human (H2B type 1-J) and *Arabidopsis* (HTB9). Histone fold segments used for structure comparison are highlighted in turquoise (human) and orange (*Arabidopsis*). Serine residue S57 (human) or S82 (*Arabidopsis*) is highlighted in dark blue. (**B**) Structural alignment of the core region of H2B from human (Turquoise) and *Arabidopsis* (orange). The protein structure of the histone fold segment of HTB9 was modeled in SWISS-MODEL (https://swissmodel.expasy.org/) using human H2B (PDB ID: 3AV1) as a template, and the predicted structure was aligned with the *Arabidopsis* HTB9 using RCSB PDB Protein Comparison Tool (http://www.rcsb.org/pdb/workbench/workbench.do?action=menu). The position of serine S57/S82 is indicated at the beginning of alpha helix-2 (α2). (**C**) Crystal structure of human nucleosome core particle (PDB ID: 3AV1) [35]. A close-up indicates an interaction of H2B with Phosphate group of the DNA via hydrogen bonds made from main-chain amide nitrogen atom (yellow arrow) as well as side-chain hydroxyl oxygen atom (black arrow) of Serine S57. Note, H2B is shown in pink, H2A in purple and the DNA phosphate backbone is shown in red. Loop 1 (L1) and α helix 2 (α2) are marked.

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
