# Peer review of "Plant Histone HTB (H2B) Variants in Regulating Chromatin Structure and Function"

_plants, 2020, doi:10.3390/plants9111435_

Round 1
Reviewer 1 Report
In this review article, Khadka and colleagues discuss the role of H2B histone variants concentrating on plant variants.
Given the nature of this article, a review article, I was surprised to find as much original, yet unpublished data in this article including a real Materials and Methods section and I am not sure about the added value that these data bring to the manuscript.
Concerning the review I have several major points:
Please do not use the term "isoforms" instead of variants. This will lead to confusion, which is already the case with researchers outside the field of histone variants. Protein isoforms derive from different splice variants of the same gene. In case of Arabidopsis histone variants, each variant is encoded by its own gene.
Concerning the nomenclature: Talbert et al had presented a terminology for histone variants, which has been adapted by most of the researchers in the field: e.g. use H2B.5 as protein name and not HTB5.
Throughout the review both terminologies are used: title: Plant H2B.s, later in the text: ...of Arabidopsis HTB variants.
Introduction: H2A.X defines regions of damaged DNA. This is not correct. H2A.X is incorporated throughout the genome (see also Yelagandula, Cell, 2014), but is phosphorylated upon DNA damage. It is its phosphorylated form, which marks damaged DNA.
When discussing the mgH2B.in in Lilium davidii (page 3), the authors give the impression that this expression pattern in pollen is unique to the lilium variant, however, Jiang et al, Plos Genetics, 2020; showed as well that the Arabidopsis H2B.8 variant is localized in sperm cell nuclei of pollen.
Concerning the expression patterns of the H2B.8 variant (page 5). This part is a little confusing. I understand that the authors want to emphasize that H2B.8 (and their orthologues in other species) are not solely expressed in seeds. However, from available RNA-seq data and as well as FP browser data available to the reviewer, the expression of H2B.8 in dry seeds (the embryo) are predominant, which has also been shown by Jiang et al with a H2B.8-RFP fusion protein.
The authors should revise this part for clarity: e.g. they say "while HTB.8 is strongly expressed in pollen and sperm cells, it is mildly expressed in seeds..." and then later: "AtH2TB.8 showed expression in mature seeds".
At several instances during the review (paragraph 3 and 4), the authors discuss mono-ubiquitination. This might be combined in a single paragraph to avoid repetition on the role of this modification in gene transcription.
Concerning the expression of H2B.5-GFP under control of the strong viral 35S promoter: I am not convinced that the localization of the H2B.5-GFP in the nucleolus is meaningful. Often proteins that over-accumulate and are not properly incorporated do accumulate in the nucleolus. How many different independent T-DNA insertion lines have been analyzed? Is there any correlation between gene expression (at the RNA level) and the accumulation in the nucleolus? Same for Figure 3: How many independent lines have been used? No information is given in the Materials and Methods section, which generation of transgenic plants has been used (T1, T2?).
The pull-down data with the 11 aa motif are quite interesting and intriguing, but I wonder whether they should not have been brought forward in a different article presenting original data and not in this review.
Page 6: when discussing the role of Rad6 in yeast, it might be useful to describe directly afterwards, which enzymes are involved in ubiquitination (HUB1/2) or deubiquitination in plants and then only later link to their role in flowering time regulation.
Jiang et al have also described the differential enrichment of H2B variants along the genome derived by ChIP-seq experiments. I do not see this information discussed in this review.
Concluding remarks: ".. which appear to be involved in establishing permissive and restrictive chromatin conformation, respectively." I do not think that there is enough evidence for such a statement, at least none published.
The first paragraph of the "concluding remarks" gives very detailed information to differences in amino acids between the different variants. This part might be better included in paragraph 2 or 3.
Minor points:
In general the manuscript should be thoroughly checked for typos and grammatical errors: e.g. first sentence of paragraph 2 (Plant H2Bs).
Arabidopsis genes should be written in italics: e.g. in introduction: HTA3 and HTA5
page 3: post-translational modifications not post-translation modifications
page 3: HRB7
page 6: impairing with mUb ?
A sentence in the Abstract (starting with ... histone variants might have specialized functions) is repeated nearly identically in the Introduction.
Author Response
Thanks for your thoughtful comments, which amended the article profoundly. We have address them all and responded point by point to all comments/suggestions in the accompanying document.

Reviewer 2 Report
I have now carefully read the manuscript presented by Khadka, Pesok and Grafi and I do not have any major issues with the paper - the subject of histones variants in plants in important and interesting. Moreover, the paper is well documented, properly written and not overload with irrelevant information.
Author Response
No comment!

Reviewer 3 Report
Brief Summary
The review entitled “Plant H2B Variants in Regulating Chromatin Structure and Function: Highlighting Arabidopsis HTB5” submitted by J. Khadka and colleagues presents on one hand the current knowledge on plant H2B variants in terms of phylogeny, post-translational modifications, pattern of expression and on the other hand some original research work on the HTB5 variant and on a specific peptide absent from this variant aiming at understanding its specificity.
Broad comments
The first part of the review provides a comprehensive understanding of the current knowledge on the structures and functions of plant H2B variants. It is timely and highly significant in the field as plant H2B and H4 variants are well less understood than H3 or H2A variants. The second part presenting experimental results on one of the Arabidopsis H2B variants (HTB5) represents to my opinion two major issues: first it hardly has its place in a review article and second the obtained results are too weak to support the strong conclusions drawn from them. This current hybrid format is in my opinion detrimental for both the review part and for the research part of the current manuscript.
I would suggest to remove the original research work to make this manuscript a sole review and reserve it to make a future research article with more robust data.
For this review manuscript, I have four major comments:
- shortening the section 4 “H2B monoubiquitination and flowering time” in a few sentences focusing and move it in section 3 “HTB variants post-translational modifications and expression pattern in Arabidopsis”.
- Split Section 3 “HTB variants post-translational modifications and expression pattern in Arabidopsis” into 2 different parts, one on PTMs and one on expression. And the part on expression would benefit in including more detailed comments on reproductive tissue expression data from Jiang D, Borg M, Lorković ZJ, Montgomery SA, Osakabe A, et al. (2020) The evolution and functional divergence of the histone H2B family in plants. PLOS Genetics 16(7): e1008964. https://doi.org/10.1371/journal.pgen.1008964 that details expression of the 11 variants in sperm cells and vegetative cells, egg and central cells.
- Add a section on the nuclear localization and genomic distribution of H2B variants, notably stemming from this recent report by Jiang et al. that provides ChIP-seq data of class I H2Bs: H2B.1, 2, 3 and 4-9-11 with endogenous antibodies.
- I would rephrase (or even withdraw if the original research work is removed from the manuscript) some of the concluding remarks (section 7.) that are hypotheses but not yet fully demonstrated facts.
Specific comments
Page 2:
Line 2/3: “suggest that the histone variants might have specialized functions in regulating chromatin structure and function, which are currently largely unknown”. Largely unknown is too strong considering the level of knowledge on plant H3.1, H3.3, cenH3 and H2A.z and H2A.w. In that respect, the following sentences should also include a comment on H3.1/H3.3 knowledge in addition of cenH3.
Line 7-11: The SQEF motif is also found in H2A.w7 (Zdravko J. Lorković, Chulmin Park, Malgorzata Goiser, Danhua Jiang, Marie-Therese Kurzbauer, Peter Schlögelhofer, Frédéric Berger, Compartmentalization of DNA Damage Response between Heterochromatin and Euchromatin Is Mediated by Distinct H2A Histone Variants, Current Biology, https://doi.org/10.1016/j.cub.2017.03.002)
The paragraph on human and mice testis specific H2Bs is rather long, the sentences on H2BL1, hTSH2B and H2BFWT could be removed to keep the focus solely on TH2B.
Page 3:
When stipulating that HTB8 molecular mass is 27 kDa, it would be good to have the average size of the other HTBs alongside for comparison.
To support the comparisons of the HTB8 C-terminal domain sequence with human and mouse TH2Bs and also with Lilium mgH2B, it would be interesting to see both alignments as a supplemental figure showing how HTB8 specific amino acids align as compared toone common with the other Arabidopsis HTBs.
Page 4 - Figure 1:
The borders of the C- and N-terminal, the histone fold, the 11aa, 21aa motif and the amino acids undergoing post-translational modifications could be highlighted on the alignment presented in 1A.
Page 5:
Line 5: I did not understand the sentence “Intriguingly, some amino acids within the motifs, which are absent in Class II variants (KPVEEKSKAE), are not modified (see Fig. 1C).”, as it seems that Lysine 11 falls within this motif and why it is intriguing.
Lines 9-10: “while deubiquitination of H2B is required for establishment of heterochromatic silencing [37]” is omitting that H2B de-ubiquitination is also associated to RNA Pol II transcription, see for example:
- Henry KW, Wyce A, Lo WS, Duggan LJ, Emre NC, Kao CF, Pillus L, Shilatifard A, Osley MA, Berger SL. 2003. Transcriptional activation via sequential histone H2B ubiquitylation and deubiquitylation, mediated by SAGA- associated Ubp8. Genes & Development 17:2648–2663. DOI: https://doi.org/10.1101/gad.1144003
- and in Arabidopsis: eLife 2018;7:e37892 doi: 10.7554/eLife.37892
I did not understand to what molecular function the Agrobacterium transformation low efficiency phenotypes of some of the H2B mutants were pointing to. And why it was in the section 3 about H2B post-translational modifications.
The sentence “First, the absence of class II variants in the euchromatic fraction reported by Bergmüller et al. [31] may implicate them in establishing and maintaining heterochromatin.” should be strongly nuanced because, to my understanding, the peptides retrieved in Bergmüller et al. after the trypsin digestion are all located in the N-terminal region and contain the amino acids that are absent in class II H2Bs. So, the lack of class II H2Bs in their mass spectrometry data rather likely comes from the technical incapability of retrieving specific peptides from this class rather than the absence of these histones in euchromatin.
“showing that class II HTB5, HTB6 and HTB7 but not HTB8 and HTB10 are essentially expressed in most tissues and organs where class I H2Bs are expressed [12].”: misleading as HTB8 is class III not II.
I did not understand the sentence “The exception was the exclusive expression of class II/III H2Bs in sperm cells and pollen”.
In the last paragraph it is not clear that H2B.S corresponds to the authors’ class III.
Page 6:
Line 5: Here also it should be stated that H2B de-ubiquitination also serves transcription and not only heterochromatinization.
To fully cite the literature, open chromatin conformation in protoplasts should also refer to “Federico Tessadori, Marie-Christine Chupeau, Yves Chupeau, Marijn Knip, Sophie Germann, Roel van Driel, Paul Fransz, Valérie Gaudin, Large-scale dissociation and sequential reassembly of pericentric heterochromatin in dedifferentiated Arabidopsis cells, Journal of Cell Science (2007) 120: 1200-1208; doi: 10.1242/jcs.000026”.
I am skeptical about drawing too much conclusions on HTB5 subnuclear localization from 35S overexpressing lines.
Page 9:
Line 4: the role H2Bub in H3K4 and 76/79 trimethylation has already been stated in the introduction and does not hold in Arabidopsis where there is no H3K76me3 and there is no major crosstalk between H2Bub and H3K4me3 (as previously mentioned in this manuscript citing Fiorucci et al.).
Line 6-7: When stating that AtCDC48 complements the yeast mutant, it should be specified that it regards cell growth but has not been tested towards H2Bub levels (if this is correct).
Page 10:
Line 1-3: “Class I and Classes II/III, which appear to be involved in establishing permissive and restrictive chromatin conformation, respectively” should be removed as no functional assays to date have shown that class I establishes permissive chromatin while classII/III establishes repressive chromatin.
Line 10-33: The large paragraph on the amino acid differences between class I and class II H2B sequences should be moved to the main text in section 2 or 3 and could be supported by a dedicated figure or supplemental figure.
Line 34-35: I did not find in the manuscript what is the 21aa motif.
Line 37-41: “We suggest that the 11/21 aa motifs, which are deficient in class II HTBs, are crucial for the function of class I H2Bs in setting up open chromatin conformation and gene transcription. This is mediated, at least partly, via the recruitment of protein complexes containing CDC48 and probably other chromatin modifiers to allow class I H2B to undergo mUb.”
This statement is absolutely not demonstrated neither in the literature nor in the presented data.
Line 41-43: “This in turn induces further modification by a COMPASS-like complex (i.e., methylation of H3K4/H3K79) and by CSP2 (unwinding of DNA duplex) leading to open, transcriptionally competent chromatin configuration.”
Again, the role H2Bub in H3K4 and 79 trimethylation does not hold in Arabidopsis where there is no H3K79me3 and there is no major crosstalk between H2Bub and H3K4me3 (as previously mentioned in the manuscript from Fiorucci et al.). Second it should be rephrased in a more hypothetical way as, first, a role of CSP2 in plant transcription and chromatin has not been reported and second the interaction with the 11 aa peptide found by GST pull-down is not enough to conclude on an interaction between class I H2Bs and CSP2.
“Based on the delay in flowering demonstrated in htb5 mutant, we propose that class II HTB5 is replacing canonical class I H2Bs at specific loci including FLC, a gene whose product plays a key role in repressing flowering in the vernalization pathway [93], to bring about transcriptional repression of flowering inhibitory genes and consequently to flowering.”
This last sentence about HTB5 role at FLC should also be rephrased in a more hypothetical manner as no experiment to test it was done.
Author Response
Many thanks for all comments and suggestions which surely improved profoundly the Article.
We responded to all comments in the attached file.
Thanks

Reviewer 4 Report
In the review manuscript "Plant H2B Variants in Regulating Chromatin Structure and Function: Highlighting Arabidopsis HTB5", the authors summarized existing data on a group of core histone H2B and discussed the potential function of H2B variants in regulating chromatin structure and function in Arabidopsis with a special focus on HTB5.
At first glance, the manuscript is interesting. However, after online search, I found the master thesis(http://aranne5.bgu.ac.il/others/PesokAnat.pdf) of Anat Pesok, who is one of the co-authors. The title of the thesis is "The role of HTB5 - a histone H2B variant - in plant development and stress-induced dedifferentiation". It seems that the new data used in the manuscript are from the thesis, but the authors did not mention/cite the thesis.
major point:
1, As I said above, it seems that the new data used in the manuscript is from the thesis. Some figures are identical: Fig 1C is the same as Fig 6 of thesis; Fig 1G is the same as Fig 11E of the thesis; Fig 3 is the same as Fig 15 of the thesis.
Fig 2A is similar to Fig 1 of the thesis.
Besides the data, the authors should pay attention to the statement in the manuscript to avoid duplication.
2, "Keywords" and "Author Contributions" are missing.
3, Why the authors focused on HTB5? As HTB5 and HTB6 are both in class II-B, the authors did not give reasons why they focused on HTB5 but not HTB6.
minor point:
1, For Fig 2A, the current display will mislead the readers that HTB5 is in "Class I" and "Class II/III" both.
2, All of the "Arabidopsis" should be italic in the manuscript.
3, "Blast" should be "BLAST".
Author Response
Thanks for your comments helping to improve the manuscript.

Round 2
Reviewer 1 Report
The revised version is much improved in clarity and the content of unpublished original data strongly reduced.
Nevertheless, the following comments should still be addressed:
Line 35-37: Please rephrase: This sentence is confusing as if DNA can also be acetylated / ubiquitinylated.
Line 52: HTR12 should be in italic.
Line 59-60... particularly within euchromatin, while phosphorylated H2A.W.7 in heterochromatin. Something is missing in this sentence.
Line 75: Is [16] the correct reference? The text mentions histone variants in humans, but the reference is a review entitled "Canonical and histone variants in plants".
Line 117: remove s on regions
Line 210: facilitates
Line 277: controls
Line 290/91: what is the reference for the absence of K72 acetylation in Class II H2B histones?
Line 313-319: I think the authors are referring here to expression data (transcriptome). Please make this clear by referring to gene expression, if no information on proteins is available here.
Line 327 / 328: use italics for HTBs and HTB8.
Lines 334 and following: I am still not convinced on the almost equal expression of H2B.8 in pollen and seeds. In Reference [15], Figure 1c (Qualitative mass spectrometry analysis of Arabidopsis H2Bs) reveales about twice as many peptides corresponding to H2B.8 in seeds compared to pollen. Furthermore the expression data in [15] Figure 1G, H show that the H2B.8 orthologues in tomato and rice are not expressed in pollen. The authors may want to turn down a little their statements.
Line 329 and following on this page: the authors refer to ref [14], it should be ref [15].
Line 398 and following: the authors refer to [70] for the localization of HTB5-GFP in nucleus and nucleolus. I cannot find this in this paper on Lili histones. If these are unpublished data from your laboratory, please indicate as such. There might be a problem with the references. Maybe you were referring to reference 77 ?
Line 4718: rich in
The authors might also want to refer to the following article: " BRHIS1 suppresses rice innate immunity through binding to monoubiquitinated H2A and H2B variants, Li et al., 2015", showing that rice SWI/SNF2 ATPase BRHIS1 interacts specifically with mono-ubiquitinated H2B.7 and not other H2B variants.
Please check references throughout the text, some are not at the correct place.
Author Response
Reviewer 1 / 2nd
The revised version is much improved in clarity and the content of unpublished original data strongly reduced.
Nevertheless, the following comments should still be addressed:
Line 35-37: Please rephrase: This sentence is confusing as if DNA can also be acetylated / ubiquitinylated.
Response: Sentence was rephrased (lines 32-34)
Line 52: HTR12 should be in italic.
Response: HTR12 is now in italics. (line 48).
Line 59-60... particularly within euchromatin, while phosphorylated H2A.W.7 in heterochromatin. Something is missing in this sentence.
Response: sentence was rephrased for clarity. (line 58).
Line 75: Is [16] the correct reference? The text mentions histone variants in humans, but the reference is a review entitled "Canonical and histone variants in plants".
Response: In the revised article reference 17 is now the correct one.
Line 117: remove s on regions
Response: corrected to region. (line 116).
Line 210: facilitates
Response: changed to facilitates (line 185)
Line 277: controls
Response: changed to controls (line 266).
Line 290/91: what is the reference for the absence of K72 acetylation in Class II H2B histones?
Response: The reference is the Plant PTM viewer and it is given in the legend to supplementary Fig. S4 and in the text (line 215).
Line 313-319: I think the authors are referring here to expression data (transcriptome). Please make this clear by referring to gene expression, if no information on proteins is available here.
Response: we clarified that the data refer to gene expression (line 315).
Line 327 / 328: use italics for HTBs and HTB8.
Response: italics done (line 316).
Lines 334 and following: I am still not convinced on the almost equal expression of H2B.8 in pollen and seeds. In Reference [15], Figure 1c (Qualitative mass spectrometry analysis of Arabidopsis H2Bs) reveales about twice as many peptides corresponding to H2B.8 in seeds compared to pollen. Furthermore the expression data in [15] Figure 1G, H show that the H2B.8 orthologues in tomato and rice are not expressed in pollen. The authors may want to turn down a little their statements.
Response: Considering all data it is quite clear that HTB8 expression is not restricted to seeds.
Line 329 and following on this page: the authors refer to ref [14], it should be ref [15].
Response: It is now in the revised manuscript reference 15.
Line 398 and following: the authors refer to [70] for the localization of HTB5-GFP in nucleus and nucleolus. I cannot find this in this paper on Lili histones. If these are unpublished data from your laboratory, please indicate as such. There might be a problem with the references. Maybe you were referring to reference 77 ?
Response: In the revised manuscript it is reference 72 (Pesok, 2014).
In this reference there is nothing about Lili histone. The reference for Lilium histone is 68 in the list.
Line 4718: rich in
Response: Changed to ‘rich in’ (line 340)
The authors might also want to refer to the following article: " BRHIS1 suppresses rice innate immunity through binding to monoubiquitinated H2A and H2B variants, Li et al., 2015", showing that rice SWI/SNF2 ATPase BRHIS1 interacts specifically with mono-ubiquitinated H2B.7 and not other H2B variants.
Response: Thanks, the Reference was added (line 234-237).
Please check references throughout the text, some are not at the correct place.
Response: references were checked and confirmed.

Reviewer 3 Report
The review manuscript and figures have been greatly improved by the authors. I still have some concerns about a few points that need to be taken into consideration before final publication.
Line 51-52: I would add that H3.3 is a replacement variant whose incorporation into chromatin is associated to transcription. This will be valuable information later to comment on the H2B variant that has been proposed to be a replacement variant as well by Jiang et al.
Line 163: Inconsistent abbreviation code: Leu Ile abbreviations are used while in the sentence before the one-letter code was chosen for amino acid abbreviation.
Line 159: The possible functional consequences of the conversion from glycine (G) 39 to acidic, negatively charged glutamic/aspartic acid (E/D) could be discussed.
Line 169 and 362: The term restrictive chromatin is unclear and not widely used, I suggest using repressive chromatin instead.
Figure 1C: I think the reference article is 64 and not 62 for the protoplast gene expression analysis.
Line 214 to 217: I am sorry to insist but while Sridhar et al. indeed show on a few loci that H2Bub deubiquitination is required for establishing silencing (and it should be noted that the extent of this mechanism has not been investigated genome wide in plants), the importance of the cycling rounds of ubiquitination de-ubiquitination that occur in the course of transcription, accompanying RNA Pol II elongation, is very well established and conserved from yeast to human (see the review by Weake and Workman 10.1016/j.molcel.2008.02.014) and Arabidopsis (See doi: 10.1016/j.jmb.2018.03.018 and doi: 10.7554/eLife.37892). Therefore, the sentence “In yeast, ubiquitination of H2B followed by deubiquitination of H2BmUb may be required for enhancement of transcription of certain genes [40].” Is highly underestimating the role of H2B deubiquitination and should be rephrased accordingly.
Line 236-238: Please also cite doi: 10.1016/j.jmb.2018.03.018 in addition to ref 54 for the description of the DUBm in Arabidopsis.
Line 251-262: This paragraph would be better positioned before the one on H2Bub, just after the description of the results in Bergmuller et al line 202-209.
Line 269-273: I still don’t see the connection between agrobacterium transformation efficiency and gene expression. Or does this refer to transgene expression?
Line 279-280: Histones are not chromatin modifiers.
Line 289-290: “Notably, while HTB8 is strongly expressed in pollen and sperm cells, it is mildly expressed in seeds, endosperm, silique and root protoplasts (Fig. 1D).” This conclusion based on the Genevestigator analysis must be reassessed. Indeed, on the shown heatmap, the color of the square represents the expression average from several subsamples. For example, for “Seed” the color is not that strong, but it is because it is an average between very strong and weaker values in the suite of experiments related to Seed. In the Genevestigator software, it is possible to click on the square and the detail of each sub-experiment appear. When doing that, I could see that the expression values in dry or early imbibed seeds is actually even higher of what is observed in pollen. In clear, when looked properly in detail, the Genevestigator data are in agreement with Jiang et al. publication and HTB8 is highly expressed in dry seeds and in pollen.
Line 314-319: Again, looking more closely in the Genevestigator database, I could find that the tomato HTB8 is very highly expressed in the seed in mature fruit, and the maize and rice HTB8s are highly expressed in the embryo in the caryopse. So, in my opinion, “characterization of the HTB8-related group as ‘seed-specific’” can be found too restrictive because HTB8 is also found in pollen but not because of the findings in tomato, maize and rice based on the Genevestigator database that have been misinterpreted.
Line 323-326: Ref 71 should refer only to the part of the sentence about the localization at NORs of the MBDs while the following part of the sentence on their role on silencing the NORs should refer to Preuss et al. 2008 DOI: 10.1016/j.molcel.2008.11.009. Also, this role seems to be established so “might be involved” could be rephrased.
Line 328-332: I would mitigate the localization at chromocenters and NORs of HTB5 first because of the overaccumulation due to the 35S promoter and second because of the lack of chromocenters in protoplast nuclei as stipulated upstream in the manuscript.
Line 339-340: Please highlight the KKRKRSRK motif in Fig. S4.
Line 348-355: This last paragraph is not about nuclear localization but genomic distribution. I would rephrase the title of this section to include both related aspects. More importantly, considering the multiple important observations made through the ChIP-seq analyses of H2B variants in Jiang et al. and how they are central to the subject of the review, I found that this paragraph is too superficial. All the conclusions of this article must be commented:
- H2B class I-A and B are everywhere along the genome
- H2B.3 is more in euchromatin and H2B.2 more in heterochromatin
- H2B.1 and 2 are found at the 5’ and 3’ end of poorly expressed genes
- H2B.3 and class I-A are enriched in gene bodies
- H2B.3 enrichment positively correlate with expression level
- H2B.3 could be a replacement variant during transcription because of the precedent statement and because its enrichment highly correlates with H3.3, it is poorly expressed in meristems but more in differentiated cells.
Line 374: “this motif is absent in class II HTB variants.” is not useful as it is stated just above (“VEEKSK motif exclusively found in class I-A HTB4/9/11”).
Line 371-377: This assumption must be removed as the original research work supporting this hypothesis is now removed.
In my opinion, adding some perspectives in the concluding remarks could be interesting, for example a comment on why we know nothing about H2B mutant phenotypes (lack of mutant lines, redundancy, lethality, no phenotypes, no previous interest?) and how this could change.
Reference 70 is hard to find online; the address of the webpage must be added in the reference table.
Author Response
Reviewer 3 2nd
Comments and Suggestions for Authors
The review manuscript and figures have been greatly improved by the authors. I still have some concerns about a few points that need to be taken into consideration before final publication.
Line 51-52: I would add that H3.3 is a replacement variant whose incorporation into chromatin is associated to transcription. This will be valuable information later to comment on the H2B variant that has been proposed to be a replacement variant as well by Jiang et al.
Response: added as suggested (line 51-53).
Line 163: Inconsistent abbreviation code: Leu Ile abbreviations are used while in the sentence before the one-letter code was chosen for amino acid abbreviation.
Response: corrected (line 161-164).
Line 159: The possible functional consequences of the conversion from glycine (G) 39 to acidic, negatively charged glutamic/aspartic acid (E/D) could be discussed.
Response: This conversion from G to E/D is discussed briefly (lines 168-170)
Line 169 and 362: The term restrictive chromatin is unclear and not widely used, I suggest using repressive chromatin instead.
Response: corrected (lines 173, 186)
Figure 1C: I think the reference article is 64 and not 62 for the protoplast gene expression analysis.
Response: Thanks this was corrected and it is now reference 66.
Line 214 to 217: I am sorry to insist but while Sridhar et al. indeed show on a few loci that H2Bub deubiquitination is required for establishing silencing (and it should be noted that the extent of this mechanism has not been investigated genome wide in plants), the importance of the cycling rounds of ubiquitination de-ubiquitination that occur in the course of transcription, accompanying RNA Pol II elongation, is very well established and conserved from yeast to human (see the review by Weake and Workman 10.1016/j.molcel.2008.02.014) and Arabidopsis (See doi: 10.1016/j.jmb.2018.03.018 and doi: 10.7554/eLife.37892). Therefore, the sentence “In yeast, ubiquitination of H2B followed by deubiquitination of H2BmUb may be required for enhancement of transcription of certain genes [40].” Is highly underestimating the role of H2B deubiquitination and should be rephrased accordingly.
Response: rephrased as suggested and indicated references were added. (Lines 233-241).
Line 236-238: Please also cite doi: 10.1016/j.jmb.2018.03.018 in addition to ref 54 for the description of the DUBm in Arabidopsis.
Response: reference was added (now references 45 and 46) Line 241.
Line 251-262: This paragraph would be better positioned before the one on H2Bub, just after the description of the results in Bergmuller et al line 202-209.
Response: done as suggested. Now lines 213-225.
Line 269-273: I still don’t see the connection between agrobacterium transformation efficiency and gene expression. Or does this refer to transgene expression?
Response: For clarity, the sentence regarding Agrobacterium was removed.
Line 279-280: Histones are not chromatin modifiers.
Response: Why not? histone variants can modify chromatin structure particularly when modified. In any case, the term was replaced with chromatin related genes. Line 280
Line 289-290: “Notably, while HTB8 is strongly expressed in pollen and sperm cells, it is mildly expressed in seeds, endosperm, silique and root protoplasts (Fig. 1D).” This conclusion based on the Genevestigator analysis must be reassessed. Indeed, on the shown heatmap, the color of the square represents the expression average from several subsamples. For example, for “Seed” the color is not that strong, but it is because it is an average between very strong and weaker values in the suite of experiments related to Seed. In the Genevestigator software, it is possible to click on the square and the detail of each sub-experiment appear. When doing that, I could see that the expression values in dry or early imbibed seeds is actually even higher of what is observed in pollen. In clear, when looked properly in detail, the Genevestigator data are in agreement with Jiang et al. publication and HTB8 is highly expressed in dry seeds and in pollen.
Line 314-319: Again, looking more closely in the Genevestigator database, I could find that the tomato HTB8 is very highly expressed in the seed in mature fruit, and the maize and rice HTB8s are highly expressed in the embryo in the caryopse. So, in my opinion, “characterization of the HTB8-related group as ‘seed-specific’” can be found too restrictive because HTB8 is also found in pollen but not because of the findings in tomato, maize and rice based on the Genevestigator database that have been misinterpreted.
Response: We rephrased this to emphasize that HTB8 is highly expressed in pollen and seeds. (lines 307-313).
Line 323-326: Ref 71 should refer only to the part of the sentence about the localization at NORs of the MBDs while the following part of the sentence on their role on silencing the NORs should refer to Preuss et al. 2008 DOI: 10.1016/j.molcel.2008.11.009. Also, this role seems to be established so “might be involved” could be rephrased.
Response: Corrected and reference was added (lines 319-320).
Line 328-332: I would mitigate the localization at chromocenters and NORs of HTB5 first because of the overaccumulation due to the 35S promoter and second because of the lack of chromocenters in protoplast nuclei as stipulated upstream in the manuscript.
Response: Chromocenters do exist they just undergo cycle of decondensation-recondensation. Here is the sentence: “The transition from leaf cells to protoplasts following treatment with cell wall degrading enzymes is accompanied by extensive changes in chromatin structure that undergo decondensation followed by recondensation of pericentric heterochromatin”.( Lines 277-280).
Quite often people reasoning accumulation in the nucleolus because of overaccumulation due to 35S promoter. Although we cannot exclude this possibility, we should take into consideration that nucleolar localization is specific. First, nucleolar localization/retention signals have been very well characterized for nuclear proteins in plants and animals. Second, If nucleolar localization resulted solely from overaccumulation (due to 35S) rather than specific targeting into the nucleolus, why many other nuclear proteins under the 35S did not show nucleolar localization?
Here are some examples:
Overproduction of NPR1-GFP in 35S::NPR1-GFP expressing plants showed no accumulation in the nucleolus (see Fig. 4A in Kinkema et al, 2000 Plant Cell 12(12): 2339–2350).
Expression of MBD proteins under the 35S promoter, in protoplasts or in transgenic plants did not show any particular nucleolar localization (See Fig. 1 and Fig. 2 in Zemach et al., 2005 The Plant Cell, Vol. 17, 1549–1558).
On the other hand, overexpression of LHP1 showed a particular localization in the nucleolus due to the presence of Nucleolar Localization signal within the hinge region; nucleolar localization was abolished upon removal of the hinge region (see Libault et al. 2005 Planta (2005) 222: 910–925; Zemach et al., 2006 The Plant Cell, Vol. 18, 133–145).
Line 339-340: Please highlight the KKRKRSRK motif in Fig. S4.
Response: the motif is indicated in Fig. S4.
Line 348-355: This last paragraph is not about nuclear localization but genomic distribution. I would rephrase the title of this section to include both related aspects. More importantly, considering the multiple important observations made through the ChIP-seq analyses of H2B variants in Jiang et al. and how they are central to the subject of the review, I found that this paragraph is too superficial. All the conclusions of this article must be commented:
- H2B class I-A and B are everywhere along the genome
- 3 is more in euchromatin and H2B.2 more in heterochromatin
- 1 and 2 are found at the 5’ and 3’ end of poorly expressed genes
- 3 and class I-A are enriched in gene bodies
- 3 enrichment positively correlate with expression level
- 3 could be a replacement variant during transcription because of the precedent statement and because its enrichment highly correlates with H3.3, it is poorly expressed in meristems but more in differentiated cells.
Response: The conclusions regarding genomic distribution of HTBs given in Jiang et al. are included.
Line 374: “this motif is absent in class II HTB variants.” is not useful as it is stated just above (“VEEKSK motif exclusively found in class I-A HTB4/9/11”).
Response: sentence was rephrased (lines 368-370)
Line 371-377: This assumption must be removed as the original research work supporting this hypothesis is now removed.
Response: Removed.
In my opinion, adding some perspectives in the concluding remarks could be interesting, for example a comment on why we know nothing about H2B mutant phenotypes (lack of mutant lines, redundancy, lethality, no phenotypes, no previous interest?) and how this could change.
Response: That would be too speculative and we prefer to avoid it.
Reference 70 is hard to find online; the address of the webpage must be added in the reference table.
Response: The web page address was added (now reference 72).

Reviewer 4 Report
The revised manuscript has satisfactorily addressed my concerns.
Author Response
Thanks, no further comments.